# Alarming Levels of Multidrug Resistance in Aerobic Gram-Negative Bacilli Isolated from the Nasopharynx of Healthy Under-Five Children in Accra, Ghana

**DOI:** 10.3390/ijerph191710927

**Published:** 2022-09-01

**Authors:** Mary-Magdalene Osei, Nicholas T. K. D. Dayie, Godfred S. K. Azaglo, Elizabeth Y. Tettey, Edmund T. Nartey, Ama P. Fenny, Marcel Manzi, Ajay M. V. Kumar, Appiah-Korang Labi, Japheth A. Opintan, Eric Sampane-Donkor

**Affiliations:** 1Department of Medical Microbiology, University of Ghana Medical School, Accra P.O. Box GP 4236, Ghana; 2FleRhoLife Research Consult, Accra P.O. Box TS 853, Ghana; 3Environmental Protection Agency, Starlet 91 Road, Ministries, Accra P.O. Box MB 326, Ghana; 4Korle-Bu Teaching Hospital, Guggisberg Avenue, Accra P.O. Box 77, Ghana; 5Centre for Tropical Clinical Pharmacology & Therapeutics, University of Ghana Medical School, Accra P.O. Box GP 4236, Ghana; 6Institute of Statistical, Social and Economic Research (ISSER), University of Ghana, Accra P.O. Box LG 25, Ghana; 7Department of Medical OCB, MSF-Belgium Headquarters, Rue de Bomel 65, 5000 Namur, Belgium; 8International Union Against Tuberculosis and Lung Disease, 2 Rue Jean Lantier, 75001 Paris, France; 9International Union Against Tuberculosis and Lung Disease, South-East Asia Office, C-6 Qutub Institutional Area, New Delhi 110016, India; 10Yenepoya Medical College, Yenepoya (Deemed to Be University) University, University Road, Deralakatte, Mangalore 575018, India; 11World Health Organization Country Office, Roman Ridge, Accra P.O. Box MB 142, Ghana

**Keywords:** Gram-negative bacilli (GNB), resistant pathogenic bacteria, nasopharyngeal carriage prevalence, extended-spectrum beta-lactamases (ESBL), AmpC, carbapenemase, Structured Operational Research and Training Initiative (SORT IT), operational research

## Abstract

Nasopharyngeal carriage of aerobic Gram-negative bacilli (GNB) may precede the development of invasive respiratory infections. We assessed the prevalence of nasopharyngeal carriage of aerobic GNB and their antimicrobial resistance patterns among healthy under-five children attending seven selected day-care centres in the Accra metropolis of the Greater Accra region of Ghana from September to December 2016. This cross-sectional study analysed a total of 410 frozen nasopharyngeal samples for GNB and antimicrobial drug resistance. The GNB prevalence was 13.9% (95% CI: 10.8–17.6%). The most common GNB were *Escherichia coli* (26.3%), *Klebsiella pneumoniae* (24.6%), and *Enterobacter cloacae* (17.5%). Resistance was most frequent for cefuroxime (73.7%), ampicillin (64.9%), and amoxicillin/clavulanic acid (59.6%). The organisms were least resistant to gentamicin (7.0%), amikacin (8.8%), and meropenem (8.8%). Multidrug resistance (MDR, being resistant to ≥3 classes of antibiotics) was observed in 66.7% (95% CI: 53.3–77.8%). Extended-spectrum beta-lactamase (ESBL)-producing bacteria constituted 17.5% (95% CI: 9.5–29.9%), AmpC-producing bacteria constituted 42.1% (95% CI: 29.8–55.5%), and carbapenemase-producing bacteria constituted 10.5% (95% CI: 4.7–21.8%) of isolates. The high levels of MDR are of great concern. These findings are useful in informing the choice of antibiotics in empiric treatment of GNB infections and call for improved infection control in day-care centres to prevent further transmission.

## 1. Introduction

Nasopharyngeal bacterial pathogens comprise a wide array of microorganisms, which include both Gram-positive bacteria and Gram-negative bacteria. These bacteria are rapidly acquired after birth and established in infancy, and they may cause infection [1,2,3]. There are two problems with nasopharyngeal carriage of pathogenic bacteria in early life. First, this may lead to the development of lower-respiratory-tract infections including pneumonia and bronchiolitis later in life [4]. It has been observed that asymptomatic nasopharyngeal carriage of pathogens is predominant in young infants and precedes the development of invasive respiratory diseases [4,5]. Second, children with nasopharyngeal carriage serve as reservoirs and transmitters of pathogens including antimicrobial resistance-producing genes [6]. Infants are vulnerable to being carriers of aerobic Gram-negative bacilli (GNB) and may be responsible for the horizontal transmission of respiratory pathogens and GNB within the community [3].

GNB are frequently encountered as causative organisms of pneumonia in children in low- and middle-income countries [4,7,8]. Colonisation of mucosal surfaces with GNB starts right after birth. Studies have shown that, by the end of first week of life, 52–83% of neonates in the neonatal intensive care unit (NICU) are colonised with GNB including ampicillin-resistant (AR) strains [8,9]. Nasopharyngeal colonization by GNB is affected by risk factors such as age, sex, acute respiratory illness, exposure to other children, socioeconomic status, antibiotic therapy, season, climate, and exposure to environmental pollutants [7,8,10].

While there are many studies on the nasopharyngeal carriage of Gram-positive bacteria (such as *Streptococcus pneumoniae* and *Staphylococcus aureus*) [11,12] and anaerobic Gram-negative bacteria (such as *Moxarella catarrhalis* and *Haemophilus influenzae*) [13,14], the evidence about aerobic GNB (as *Klebsiella pneumoniae*, *Escherichia coli*, *Acinetobacter baumanii*, and *Pseudomonas aeruginosa*) carriage is limited. Previous studies from Brazil, Indonesia, Angola, and the Netherlands reported a GNB carriage prevalence varying from 5% to 57% in healthy children [3,4,7,8,15]. There is no published evidence on the nasopharyngeal carriage of aerobic GNB from Ghana. Such information is crucial for two reasons. First, this helps in understanding the common aerobic GNB prevalent in Ghana and their potential in causing respiratory infections. Second, it serves as a baseline for the monitoring of future trends even though the result may not be relevant for clinical decision making in study participants who are asymptomatic and are carriers.

Therefore, we undertook this study among healthy under-five children attending selected day-care centres in the Accra metropolis of the Greater Accra region of Ghana from September to December 2016, to (i) determine the prevalence of nasopharyngeal colonization of GNB, and (ii) describe the common organisms isolated and their antimicrobial resistance patterns including multidrug resistance (MDR), as well as extended-spectrum beta lactamase (ESBL)-, AmpC-, and carbapenemase-producing bacilli.

## 2. Materials and Methods

### 2.1. Study Design

This was a cross-sectional study involving retrospective analysis of frozen swab samples stored at −80 °C in skim milk tryptone glucose glycerol broth (STGG). These samples were collected as part of a previous study conducted in 2016 which examined carriage rates of *S. pneumoniae* and *S. aureus* [16,17].

### 2.2. Setting and Study Sites

Ghana is a Western Africa country situated on the coast of the Gulf of Guinea [18], with a population of 30.8 million [19]. Ghana’s administrative capital is the coastal city of Accra, with a population of 16.7 million, and it has a warm and humid climate [11]. The per capita income of Ghana is 5693 USD.

The study was carried out in nurseries and kindergartens (referred to as day-care centres hereafter) within the Accra metropolis of the Greater Accra region of Ghana from September to December 2016. A list of nurseries and kindergartens in the Accra metropolis was obtained from the Ghana education service. Seven of these day-care centres were randomly selected, and written consent was obtained from the parents of the children.

The seven day-care centres spanned three districts, namely, Ashiedu Keteke sub-metropolitan for Palladium, Okaikoi South sub-metropolitan for Kaneshie, and Ablekuma South sub-metropolitan for Mamprobi and Korle-Gonno. Children whose parents did not consent and children who declined to provide assent after parental consent were excluded. Children with active upper-respiratory-tract infections or who had received antibiotics within the last 2 weeks prior to sampling were also excluded.

### 2.3. Sample Collection

One nasopharyngeal swab (NPS) was collected from each study participant using FlOQ swab sticks (Copan Flock Technologies, Brescia, Italy). The swabs were directly inoculated into vials containing skim milk tryptone glucose glycerol (STGG) transport medium (Oxoid, Basingstoke, UK) [20].

Inoculated swabs were transported to the University of Ghana Medical School, Department of Medical Microbiology, Bacteriology Unit, and were stored at −80 °C. In previous studies, *S. pneumoniae* [16] and *S*. *aureus* [17] were isolated, antimicrobial susceptibility testing and serotyping were performed, and the data were analysed using standard techniques. For the purpose of this study, GNB were isolated, and antimicrobial susceptibility testing was performed.

### 2.4. Laboratory Processing

Archived swabs in STGG stored in a −80 °C freezer were brought out to thaw to room temperature and vortexed. A drop of the sample in STGG was inoculated into sterile tryptic soy broth (Oxoid, Ltd., Basingstoke, UK) and incubated for 48 h at 35 ± 2 °C. After 48 h incubation, a loopful of the broth was plated onto sterile and dried MacConkey agar (Oxoid Ltd., Basingstoke, UK) and incubated aerobically for 18–24 h.

Identification of GNB was performed using matrix-assisted laser desorption ionization time-of-flight mass spectrometry (Bruker BD MALDI-TOF MS), after plating on nutrient agar and incubation under aerobic conditions (ambient air) at 35 ± 2 °C for 18 to 24 h.

### 2.5. Antimicrobial Susceptibility Testing

Antimicrobial susceptibility tests were performed on the GNB isolates using the Kirby–Bauer disc diffusion method, and breakpoints determined as per the Clinical Laboratory Standards Institute (CLSI) guidelines [21]. A 0.5 McFarland equivalent suspension of each isolate was inoculated on a sterile and dried Mueller–Hinton agar (MHA) (Oxoid, Hampshire, UK) plate, followed by incubation at 35 ± 2 °C for 16–18 h.

For each isolate, within 15 min following the adjustment of the turbidity of the inoculum to 0.5 McFarland suspension using a nephelometer (BD Phoenix Spec TM, Becton, Dickinson and Company, Franklin Lakes, NJ, USA), a sterile cotton swab was dipped into the adjusted suspension and evenly streaked across the entire surface of a sterile and dried Mueller–Hinton agar plate with the purpose of obtaining a semi-confluent growth post-incubation.

The antimicrobials and their various concentrations that were used for susceptibility testing were in the following classes: aminoglycosides (amikacin (30 μg) and gentamicin (10 μg)), penicillins (ampicillin (10 μg), amoxiclav (20/10 μg), and piperacillin–tazobactam (100/10 μg)), tetracylines (tetracycline (30 μg)), cephalosporins (cefoxitin (30 μg), cefepime (30 μg), cefotaxime (30 μg), ceftazidime (30 μg), ceftriazone (30 μg), cefuroxime and (30 μg)), sulfonamides (trimethoprim/sulfamethoxazole (1.25/23.75 μg)), fluoroquinolones (ciprofloxacin (5 μg)), and carbapenems (meropenem (10 μg)), all from Oxoid [22,23]. Extended-spectrum beta-lactamase (ESBL)-producing bacteria were screened with ceftazidime (10 μg) and ceftazidime/clavulanic acid (10/10 µg), AmpC-producing bacteria were screened with cefoxitin (30 μg), and carbapenemase-producing bacteria were screened with meropenem (10 μg) using the disc diffusion technique [21]. Carbapenemase-producing bacteria were confirmed using the modified Hodge test. *E. coli* American Type Culture Collection (ATCC) 25922 control strain was used as quality control for the antimicrobial resistance tests. The breakpoints for Enterobacterales in CLSI 2021 were used for the interpretation [21]. All tests were conducted at the University of Ghana Medical School, Department of Medical Microbiology, Bacteriology Unit, following quality-assurance protocols.

### 2.6. Study Population

The study population included all healthy children aged ≤ 5 years (whose parents consented) from the seven randomly selected day-care centres of the Accra metropolis of the Greater Accra region of Ghana from September to December 2016. We calculated a sample size of 323 assuming a prevalence of 30% [7], precision of 5%, and 95% confidence intervals. We included all the samples (*n* = 410) collected as part of the previous study to meet the sample size requirements.

### 2.7. Data Variables, Management, and Analysis

Key data variables included age in months, sex (male/female), district, culture result (growth/no growth), GNB culture results (positive/negative), organisms isolated, results of susceptibility testing to each drug (resistant/susceptible), multidrug resistance (yes/no), ESBL-producing bacteria (present/absent), AmpC-producing bacteria (present/absent), and carbamapenase-producing bacteria (present/absent).

Data extracted from a paper-based individual participant’s record were entered into MS Excel, before being exported to STATA software (version 14.1, StataCorp LLC, College Station, TX, USA) for analysis. Descriptive statistical analysis was performed, and frequencies and proportions were calculated. Bacterial isolates were described, and culture positivity (proportion of samples which were positive for GNB) along with 95% confidence intervals (CI) was calculated. Differences in culture positivity by age and sex were examined using the chi-square test. For each of the organisms isolated, the AMR profile (proportion resistant to each antibiotic tested) and prevalence of MDR (defined as being resistant to three or more classes of antibiotics) was calculated along with 95% CIs [24]. A *p*-value of 0.05 or below was considered statistically significant.

## 3. Results

### 3.1. Demographics of Study Population

There was a total of 410 samples collected from the healthy under-five children. Of them, 210 (51%) were males. The mean age of the children was 39 months (SD = 10.6, range 6–60 months).

### 3.2. Nasopharyngeal GNB Carriage Prevalence

The nasopharyngeal carriage prevalence of aerobic GNB was 13.9% (95% CI: 10.8–17.6%). There was no statistically significant difference in prevalence by age, sex, or district (Table 1). A total of nine bacterial species were identified, which included *E. coli* (26.3%), *K. pneumoniae* (24.6%), *E. cloacae* (17.5%), *A. baumannii* (8.9%), *P. aeruginosa* (7.0%), and *Serratia marcescens* (1.8%) (Table 2).

### 3.3. Antimicrobial Resistance Profile

Resistance was most frequently observed for cefuroxime (73.7%), followed by ampicillin (64.9%) and amoxicillin/clavulanic acid (59.6%). The organisms were least resistant to gentamicin (7.0%), followed by amikacin and meropenem (both at 8.8%) (Figure 1).

### 3.4. MDR and Phenotypic Resistance Producing Genes

Overall, MDR was observed in 66.7% (95% CI: 53.3–77.8%) of isolates. MDR varied among the individual bacteria and was relatively higher in *A. baumannii* (100%), *E. cloacae* (90%), and *E. coli* (80.0%). ESBL-producing bacteria constituted 17.5% (95% CI: 9.5–29.9%) of isolates with relatively higher prevalence in *E. coli* (33.3%), *K. pneumoniae* (21.4%), and *A. baumannii* (20.0%). AmpC-producing pathogenic bacteria constituted 42.1% (95% CI: 29.8–55.5%) of isolates, while carbapenemase-producing pathogenic bacteria constituted 10.5% (95% CI: 4.7–21.8%) of isolates (Table 3).

## 4. Discussion

This is the first study from Ghana reporting on the prevalence of nasopharyngeal carriage of aerobic GNB and their resistance patterns in healthy under-five children. While there are many studies globally on nasopharyngeal carriage of Gram-positive bacteria and anaerobic Gram-negative bacteria, the evidence on aerobic GNB is limited. Thus, this study also contributes to the limited global evidence on this issue. There were three key findings. First, one in seven children was a carrier of aerobic GNB. Second, *E. coli*, *K. pneumoniae*, and *E. cloacae* were the most common organisms and accounted for two-thirds of all organisms isolated. Third, resistance levels were high, and two-thirds of the organisms exhibited MDR. We discuss these findings below.

The prevalence of aerobic GNB carriage was 14% in our study. This was within the range of previous prevalence reported by other studies on this issue, ranging widely from as low as 5% to as high as 57%. A study conducted in 1999 reported that GNB carriage was more prevalent in Brazilian (50%) and Angolan (57%) children than in Dutch (5%) children [15]. These differences were attributed to higher family sizes (and, thus, crowding and higher chance of transmission), poor hygienic conditions, low socioeconomic status, and warmer climates in Brazil and Angola compared to the Netherlands. The situation in Ghana is more similar to Brazil and Angola than to the Netherlands. Another study from Brazil published in 2010 reported a prevalence of 9%, markedly lower than the previous study conducted in 1999 [3]. A similar study from Indonesia published in 2013 reported a prevalence of 27% [7].

The predominant organism isolated in our study was *E. coli*, followed by *K. pneumoniae* and *E. cloacae.* This is in line with the predominant organisms found in the study from Brazil, although *Enterobacter* spp. and *K. pneumoniae* were more common than *E. coli* (10%) [3]. In the study from Indonesia, *K. pneumoniae* was the most common bacterium, followed by *Enterobacter* spp. and *A*. *baumannii* complex. *E.coli* was not observed in Indonesia [7].

We found high levels of resistance to commonly used first-line antibiotics such as cefuroxime, ampicillin, and amoxicillin/clavulanic acid. The GNB were least resistant to gentamicin, amikacin, and meropenem. These findings are in line with other studies. MDR levels were high in our study at 66% compared to Brazil (24%) [3]. The study from Indonesia did not report on MDR prevalence. We reported the levels of ESBL-producing, AmpC-producing, and carbapenemase-producing aerobic GNB for the first time ever in under-five healthy children. The study from Indonesia, however, reported resistance to cefotaxime and ceftriaxone, and the presence of an ESBL was not confirmed [7].

The strengths of the current study were the large sample size, the random sampling process for selection of day-care centres, and the quality-assured microbiology procedures to identify aerobic GNB and resistance. The conduct and reporting of the study were also in line with the STROBE statement (Strengthening the Reporting of Observational Studies in Epidemiology) [25].

There were a few limitations to our study. Firstly, while we had a reasonably large sample size to estimate the overall prevalence of GNB carriage in healthy under-five children, the sample was not sufficient to estimate the resistance levels in individual bacteria. Secondly, the study was conducted in one of the areas of Accra and, thus, we feel that the findings are not generalizable nationwide. Thirdly, information on response rates and characteristics of the excluded children was not available. Hence, we are unable to comment on the impact of this on the overall estimates. Lastly, we used frozen samples collected 5 years ago, which may raise concerns about the viability of GNB and the possibility that we may have underestimated the overall prevalence. This may not be a major limitation because a study conducted by Hare et al. in 2011 concluded that studies which rely on the viability of respiratory pathogens can be conducted using original swabs stored at −70 °C for at least 12 years [20].

Despite these limitations, there are some important implications. Firstly, the study provides information on the common aerobic GNB prevalent in Ghana, their resistance patterns, and their potential in causing respiratory infections, while it also serves as a baseline for the monitoring of future trends. Even though the results may not be relevant for clinical decision making in the carriers, this provides clues about the drugs to be used in the empirical treatment of children when they develop pneumonia and other invasive infections. Secondly, since the study used samples collected in 2016, this reflects the situation 6 years ago. This warrants a follow-up study to assess the current rates of nasopharyngeal carriage and resistance levels. Thirdly, a single study from one city may not be representative of the situation in Ghana. This calls for either a nationwide study or strengthening surveillance systems to routinely capture the GNB carriage rates in healthy children. This may require setting up sentinel sites from which nasopharyngeal samples are collected on a periodic basis and analysed using molecular technology to know the circulating genes responsible for antimicrobial resistance. It may also help in setting up prospective follow-up studies to determine the factors associated with progression from carriage to infection. Lastly, the high levels of GNB carriage and MDR call for improved infection prevention and control in day-care centres to prevent any further transmission [26].

## 5. Conclusions

In this first ever study from Ghana, we found that one in seven healthy under-five children attending the day-care centres in Accra, Ghana, carry aerobic GNB in their nasopharynx. The resistance levels were high, and MDR was seen in two-thirds of the bacteria isolated. These findings are of concern. We recommend a nationwide surveillance system with data collected periodically from sentinel sites; this will help in generating nationally representative information that can be used to inform the choice of antibiotics in empiric treatment of infections caused by GNB. A follow-up study is required to know the current status of MDR in this cohort of children. The evidence also calls for better infection prevention and control at the day-care centres in Ghana to prevent further transmission.

## Figures and Tables

**Figure 1 ijerph-19-10927-f001:**
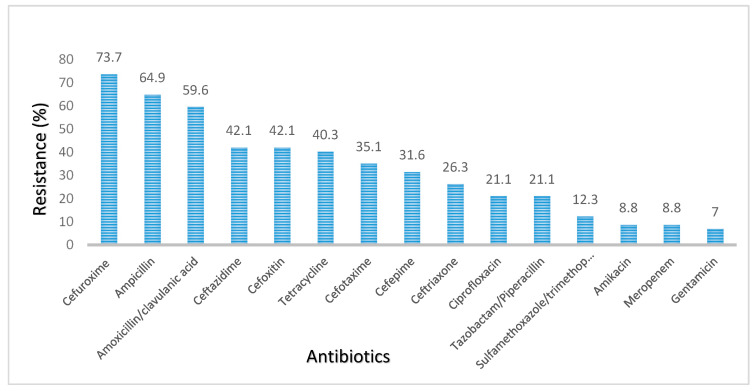
Antibiotic resistance pattern of aerobic Gram-negative bacteria isolated from the nasopharynx of healthy under-five children in Accra, Ghana, from September to December 2016.

**Table 1 ijerph-19-10927-t001:** Prevalence of aerobic Gram-negative bacteria (by age, sex, and district) isolated from the nasopharynx of healthy under-five children in Accra, Ghana, from September to December 2016.

		Number of Samples	Prevalence of GNB		*p*-Value
			N	(%)	
**Total**		410	57	(13.9)	
**Sex**	Male	210	26	(12.3)	0.36
	Female	200	31	(15.5)	
**Age (months)**	≤12	5	2	(40.0)	0.20
	13–24	56	10	(17.8)	
	25–36	172	24	(13.9)	
	37–48	129	14	(10.8)	
	49–60	48	7	(14.5)	
**District**	Ashiedu Keteke	217	34	(15.7)	0.53
	Okaikoi South	55	7	(12.7)	
	Ablekuma south	138	16	(11.6)	

**Table 2 ijerph-19-10927-t002:** Aerobic Gram-negative bacteria isolated from the nasopharynx of healthy under-five children in Accra, Ghana, from September to December 2016.

Bacteria Isolated	Number	(%)
Total isolates	57	(100)
*Escherichia coli*	15	(26.3)
*Klebsiella pneumoniae*	14	(24.6)
*Enterobacter cloacae*	10	(17.5)
*Acinetobacter baumannii*	5	(8.9)
*Pseudomonas aeruginosa*	4	(7.0)
*Erwinia species*	3	(5.3)
*Providencia rettgeri*	3	(5.3)
*Pantoea septica*	2	(3.5)
*Serratia marcescens*	1	(1.8)

**Table 3 ijerph-19-10927-t003:** Percentages of MDR and ESBL-, AmpC-, and carbapenemase-producing bacteria isolated from the nasopharynx of healthy under-five children in Accra, Ghana, from September to December 2016.

Bacteria Isolated	Number	MDR %	ESBL- Producing %	AmpC- Producing %	Carbapenemase-Producing %
Total	57	66.7	17.5	42.1	10.5
*Escherichia coli*	15	80.0	33.3	26.7	20.0
*Klebsiella pneumoniae*	14	57.1	21.4	28.6	7.1
*Enterobacter cloacae*	10	90.0	0.0	0.0	-
*Acinetobacter baumannii*	5	100.0	20.0	100.0	20.0
*Pseudomonas aeruginosa*	4	25.0	0.0	0.0	0.0
*Ewinia* species	3	0.0	0.0	0.0	-
*Providencia rettgeri*	3	66.7	33.3	33.3	-
*Pantoea septica*	2	50.0	0.0	50.0	-
*Serratia marcescens*	1	0.0	0.0	0.0	-

MDR = multidrug resistance; ESBL = extended-spectrum beta lactamase.

## Data Availability

The dataset used in this paper is available upon request.

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
