# Peer review of "Alarming Levels of Multidrug Resistance in Aerobic Gram-Negative Bacilli Isolated from the Nasopharynx of Healthy Under-Five Children in Accra, Ghana"

_ijerph, 2022, doi:10.3390/ijerph191710927_

Round 1

Reviewer 1 Report

The manuscript entitled "Alarming levels of multidrug resistance in aerobic gram-negative bacilli isolated from the nasopharynx of healthy under five children in Accra, Ghana" is of general interest to the medical community and provides insights for strengthening public health policies. It is well written and the number of samples is sufficient.

I have only a few observations and suggestions:

- On lines 148-154 a group of antibiotics is mentioned, why were these specific antibiotics chosen? please clarify

-Of the GNB isolated, which one is the most relevant to be found in the nasopharynx? A. baumanii is a very important microorganism related to sanitary infections, it could be interesting to exploit it in the discussion section.

-In Figure 1 please change under the third bar "Amoxicillin/clavulanic acd" to "Amoxicillin/clavulanic acid" in the same way, please change under the ninth bar, "Ceftrixone" to "Ceftriaxone".

-On line 239 please revise the word "and" between the bacteria names, it does not need to be italicized.

-Line 242. You mentioned that the prevalence of aerobic GNB was 14%, indicating that this is different from previous work reporting prevalences between 5% and 57%, however 14% is in the range. Please check the wording.

-Line 245. You mentioned similar studies conducted in Brazil and Angola, this supports a broader discussion such as hygienic, climatic or income conditions which could be similar to Ghana and different from the Netherlands. You might explore this suggestion.

Author Response

Author's Notes to Reviewer

Thanks much for taking time off your busy schedule to review this paper.

Number

Reviewer’s comment

Author’s response

1

On lines 148-154 a group of antibiotics is mentioned, why were these specific antibiotics chosen? please clarify

Thank you for this comment.

The antibiotics mentioned in lines 148-154, are common antibiotics recommended by the Standard Treatment Guideline from the Ministry of Health, Ghana National Drug Programme, 2017 (GNDP) and also WHO AWaRe 2021 classification of antibiotics for categories of Access, Watch and Essential Medicines for children.

The various references to the above has been added to line 154.

2

Of the GNB isolated, which one is the most relevant to be found in the nasopharynx? A. baumanii is a very important microorganism related to sanitary infections, it could be interesting to exploit it in the discussion section.

Thank you for this comment.

From literature the most relevant GNB found in the nasopharynx are that have the possibility of causing pneumonia and bronchiolitis later in life – which include Acinetobacter baumannii, E. coli, Klebsiella pneumoniae and Pseudomonas aeruginosa – line 78.

3

In Figure 1 please change under the third bar "Amoxicillin/clavulanic acd" to "Amoxicillin/clavulanic acid" in the same way, please change under the ninth bar, "Ceftrixone" to "Ceftriaxone".

Thank you, we have made the changes.

4

On line 239 please revise the word "and" between the bacteria names, it does not need to be italicized.

Thank you, we have made the changes.

5

Line 242. You mentioned that the prevalence of aerobic GNB was 14%, indicating that this is different from previous work reporting prevalences between 5% and 57%, however 14% is in the range. Please check the wording.

Thank you, we have rephrased the sentence.

6

Line 245. You mentioned similar studies conducted in Brazil and Angola, this supports a broader discussion such as hygienic, climatic or income conditions which could be similar to Ghana and different from the Netherlands. You might explore this suggestion.

Line 245. The suggestion made was excellent hence, that has been incorporated in the discussion. This is because most of the conditions in Brazil and Angola are similar to that in Ghana.

Reviewer 2 Report

Main comments:

The manuscript titled “Alarming levels of Multidrug Resistance in aerobic Gram-negative bacilli isolated from the nasopharynx of healthy under-five children in Accra, Ghana” has identified various Nasopharyngeal aerobic Gram-negative bacilli (GNB) and it’s antibiotics resistance.

1.      There are many major concerns on this work. Why author publishing the sample data from 2016 in 2022? Author should test the latest Nasopharyngaeal sample to know the recent antibiotics resistance population in Ghana to have a correct data and its future treatment options.

2.      The Disc experiments using in this manuscript are not deep enough to have conclusion on Antibiotic resistance population. Author need to support the data with providing a strong experiment.

3.      A proper control was not mentioned in the manuscript nor in the material and method sections.

4.      Author did not provide the number of passage used for the bacterial population to arrive for the conclusion of defining it as “Antibiotic resistance strains”.

5.      With the lack of strong experiments and samples age, the manuscript is not a right fit for this Journal.

Author Response

* Author's Notes to Reviewer

Thank you very much for your comments. They are very much acknowledged and the following response made to address them

Number

Reviewer’s comments

Authors response

1

There are many major concerns on this work. Why author publishing the sample data from 2016 in 2022? Author should test the latest Nasopharyngeal sample to know the recent antibiotics resistance population in Ghana to have a correct data and its future treatment options.

Thank you for the comment.

We partially agree with the view that it would have been much better if we had collected and presented more recent data.

However, we think that the data presented in the paper (even though not recent), is very much relevant. Given that there has not been any evidence on the issue of aerobic GNB nasopharyngeal carriage in Ghana, this study adds to the evidence base and provides a baseline against which we can compare the findings of future studies.

2

The Disc experiments using in this manuscript are not deep enough to have conclusion on Antibiotic resistance population. Author need to support the data with providing a strong experiment.

Thank you. We are unclear of the term ‘not deep enough’.

All the antimicrobial susceptibility tests were performed on the GNB isolates using the Kirby Bauer’s disc diffusion method and breakpoints determined as per the 2021 Clinical Laboratory Standards Institute (CLSI) guidelines.

All tests were conducted by qualified and trained staff at the University of Ghana Medical School, Department of Medical Microbiology, Bacteriology Unit, following quality-assurance protocols.

3

A proper control was not mentioned in the manuscript nor in the material and method sections.

Thank you for this comment.

Kindly refer to the section found in the materials and methods section. Line 126-164. We have clearly described the controls used in the testing process. E. coli American Type Culture Collection (ATCTC) 25922 control strain was used as quality control for the antimicrobial resistance tests.

4.

Author did not provide the number of passage used for the bacterial population to arrive for the conclusion of defining it as “Antibiotic resistance strains”.

Thank you. We are unclear what ”passage” in the comment means.

All the antimicrobial susceptibility tests were performed on the GNB isolates using the Kirby Bauer’s disc diffusion method and breakpoints determined as per the 2021 Clinical Laboratory Standards Institute (CLSI) guidelines.

AmpC-producing bacteria was tested using cefoxitin (30 μg) disc. Line 156-157.

ESBL-producing bacteria was tested with the Double dics combination using ceftazidime (10 μg) and ceftazidime/clavulanic acid (10/10 µg). Line 155-156.

Carbapenemases-producing bacteria were screened with meropenem (10 µg) and then confirmed using Modified Hodge Test. Line 157-159

5

With the lack of strong experiments and samples age, the manuscript is not a right fit for this Journal.

Thank you.

We beg to disagree with the comment. We have followed the standard procedures as per CLSI guidelines. The concern of age of the frozen samples and viability of GNB is slightly misplaced. A study conducted by Hare et al., in 2011 concluded that studies which rely on the viability of respiratory pathogens can be conducted using original swabs stored at -70°C for at least 12 years. This has already been discussed in the paper in Line 282.

Round 2

Reviewer 2 Report

Authors has revised and answered the given comments